# Prospects of Novel and Repurposed Immunomodulatory Drugs against Acute Respiratory Distress Syndrome (ARDS) Associated with COVID-19 Disease

**DOI:** 10.3390/jpm13040664

**Published:** 2023-04-13

**Authors:** Smruti Sudha Nayak, Akshayata Naidu, Sajitha Lulu Sudhakaran, Sundararajan Vino, Gurudeeban Selvaraj

**Affiliations:** 1Department of Bio-Sciences, School of Bio Sciences and Technology, Vellore Institute of Technology, Vellore 632014, Tamil Nadu, India; 2Department of Biotechnology, School of Bio Sciences and Technology, Vellore Institute of Technology, Vellore 632014, Tamil Nadu, India; 3Centre for Research in Molecular Modeling, Department of Chemistry and Biochemistry, Concordia University-Loyola Campus, Montreal, QC H4B 1R6, Canada

**Keywords:** COVID-19, ARDS, clinical trials, drug discovery, drug repurposing, machine learning, deep learning, network medicine, drug–target network

## Abstract

Acute respiratory distress syndrome (ARDS) is intricately linked with SARS-CoV-2-associated disease severity and mortality, especially in patients with co-morbidities. Lung tissue injury caused as a consequence of ARDS leads to fluid build-up in the alveolar sacs, which in turn affects oxygen supply from the capillaries. ARDS is a result of a hyperinflammatory, non-specific local immune response (cytokine storm), which is aggravated as the virus evades and meddles with protective anti-viral innate immune responses. Treatment and management of ARDS remain a major challenge, first, because the condition develops as the virus keeps replicating and, therefore, immunomodulatory drugs are required to be used with caution. Second, the hyperinflammatory responses observed during ARDS are quite heterogeneous and dependent on the stage of the disease and the clinical history of the patients. In this review, we present different anti-rheumatic drugs, natural compounds, monoclonal antibodies, and RNA therapeutics and discuss their application in the management of ARDS. We also discuss on the suitability of each of these drug classes at different stages of the disease. In the last section, we discuss the potential applications of advanced computational approaches in identifying reliable drug targets and in screening out credible lead compounds against ARDS.

## 1. Introduction

SARS-CoV-2 is a member of the Coronaviridae family with a unique positive sense RNA as its genome. Clinical manifestations of the infection can range from the common cold to acute pneumonia, and in critical cases from respiratory distress to septic shock. Acute respiratory distress syndrome (ARDS), a life-threatening condition attributed to respiratory failure, is prevalent in COVID-19-infected patients with co-morbidities (diabetes, hypertension, autoimmune disorders, old age) and is a major cause of COVID-19-associated mortality (L. Tan, 2020) [1]. According to WHO, globally over 500 million cases have been reported and around 6 million COVID-19-related deaths have occurred as of January 2023 [2]. COVID-19 has posed an unprecedented challenge to many healthcare systems worldwide owing to its efficient person-to-person transmission, associated risks of severe complications that dictate hospitalization, and lack of specific treatment options for disease management [3].

Examination of lung tissues of deceased patients as a result of COVID-19 has revealed alveolar damage, fibrin storage, and small and large pneumonic vessels, which can be directly attributed to ARDS (D Marotto, 2020) [4] (Schett et al., 2020) [5]. In ARDS, activated immune cells produce a variety of mediators including pro-inflammatory cytokines, neutrophil myeloperoxidases, and other proteolytic enzymes, which cause massive tissue destruction and can lead to respiratory disabilities (D Marotto, 2020) [4]. Treatment and management of ARDS remain a major challenge, particularly in SARS-CoV-2-infected patients with co-morbidities. For the current study, we review the pathogenesis of SARS-CoV-2 and the hyperinflammatory response unleashed as a result of infection (which leads to ARDS). Then we present the current state of interventions available to modulate immune responses to manage ARDS. Lastly, we discuss key tools and methods in the field of systems and computational biology, including ML, which are being employed or can be employed for new drug discovery for immune modulation specifically against the COVID-19 disease.

## 2. Pathogenesis of COVID-19 Disease

### 2.1. SARS-CoV-2 Structural Components

SARS-CoV-2 is a positive-stranded RNA virus that contains (a) structural proteins, like the spike proteins (S) that bind to the host cell receptor; the nucleocapsid (N), which secures hereditary infection data; the matrix (M); and the envelope (E); (b) Non-structural proteins including proteases (nsp3, nsp5) and RdRp (nsp12) (M Levi, 2020) [6]. Proteases are an ideal target for intervention as they support viral replication and hence are integral to virus survival (Schett et al., 2020) [5]. Viral transmembrane spike (S) glycoprotein is a multi-subunit protein that promotes the cellular transmission of SARS-CoV-2 and can be catalytically cleaved to subunits S1 and S2 [7]. While the S2 subunit directly binds to the ACE2 receptor, the S1 subunit acts as a facilitator (Figure 1).

### 2.2. Immune Dysregulation during COVID-19—ARDS

SARS-CoV-2 enters the respiratory tract through microdroplets and invades the smooth muscle cells and the endothelial cells present in the respiratory tract. After reaching the lungs, the virus infects the alveolar cells, which consist of type 1 and type 2 pneumocytes and alveolar macrophages where the virus starts replicating [8]. The protective immune response to the virus is mainly dominated by the release of interferon II and III. Because of the evasion strategies of the virus, ideal anti-viral responses are arrested/blocked, which in turn unleashes a cascade of hyperinflammatory responses, majorly mediated by TNF-alpha and IL-6 cytokines in the alveolar tissue as well as in the bronchial epithelial cells, which leads to lung tissue injury and damage [9]. Susceptible patients have been observed to have high levels of inflammatory cytokines including IL-1beta, IL-2, and IL-10 (apart from TNF-alpha and IL-6) in both sera and in autoptic tissues. As a result, lung tissues of patients suffering from the severe disease have been observed to have infiltration of inflammatory leukocytes and lymphocytes, which were associated with fatal tissue injury and damage caused as a result of exacerbated innate immune responses (refer to Figure 2 and Figure 3). As a consequence of tissue damage, a condition develops that is clinically manifested as dyspnea and tachypnea (which involves the need for heavy breathing).

Interestingly, patients with COVID-19 have also been reported to generate autoantibodies against specific and immunomodulatory cytokines that would otherwise have balanced the hyper-inflammatory responses [10,11]. On the other hand, a cohort study from New York validated IL-6 and TNF-alpha as designated biomarkers of disease severity by using a multiplex cytokine profiling assay on the serum samples. The authors also suggested these cytokines as potential targets of intervention along with other molecules involved in either the upstream or downstream signaling pathways associated with the two cytokines [12]. Figure 2 presents protein–protein interaction (PPI) networks associated with sepsis and with COVID-19-induced ARDS, highlighting differential profiles of hyperinflammation in the two diseases. Figure 4 illustrates the molecular mediators/signaling pathways involved in the exacerbated innate immune response associated with ARDS as compared to protective innate immune responses against viral infections.

#### Neutrophils and Coronavirus (COVID-19)

Neutrophils are white blood cells (WBCs) that circulate in the bloodstream and become active upon any signal of invasion or intrusion [13]. Neutrophils’ fundamental function is the phagocytosis of invading bacteria. They also have a role in the defense against viruses as they attack the infected cells. The most important defense mechanism adopted by these leukocytes includes the formation and release of neutrophil extracellular traps (NETs) and the release of inflammatory cytokines and reactive oxygen species (ROSs). Please refer to Figure 3, which illustrates the role of neutrophils in lung tissue damage during ARDS. In many ARDS-related infections of the lung, such as influenza and against other coronaviruses, neutrophils play a significant protective role [14,15].

A multi-omics study revealed a significant association of ARDS (induced by SARS-CoV infection) with neutrophil infiltration and degranulation (of tertiary level granules) based on the analysis of acquired proteomic data from the affected patients [16]. Neutrophils can cause lung tissue damage (associated with ARDS) by the (i) overactivation of NADPH oxidase, which results in the release of massive quantities of ROS in the alveolar tissue, causing inflammation injury; (ii) release of tertiary granules that contain a high proportion of metalloprotease, which can damage the extracellular matrix in the local tissues during an execrated response; (iii) release of dysregulated NETs, as an excessive quantity of NETs has been linked with COVID-19 disease severity and mortality. The primary reason for tissue damage because of NET release is the resultant thrombosis in tissue capillaries, which in turn results in fibrin deposition and hence a fatal shortage in oxygen supply in lung tissues [17].

### 2.3. COVID-19 in Patients with Autoimmune Diseases

The dynamics of infections and autoimmune conditions largely remain unknown. For example, rheumatic arthritis (RA) is associated with persistent infections, which can be interpreted in two different ways. From one perspective, one can say that the locally execrated immune response in RA might directly facilitate the colonization of joints by the microbe, or from the other perspective, the deviant immune response generated by the host to an infection might lead to intense and consistent joint pain [18]. A Korean investigation of patients with autoimmune diseases found that patients with rheumatic diseases were more likely to acquire SARS-CoV-19 infection, develop severe disease, and had a high rate of mortality compared to patients without RA [19]. Another multicentric study from Spain and USA confirmed a greater chance of hospitalization in patients with autoimmune diseases when infected with SARS-CoV-2 as compared to patients with influenza, emphasizing that patients with autoimmune diseases are specifically susceptible to COVID-19 infection [20].

On the other hand, infection with SARS-CoV-2 has also been linked with the induction of local and systemic autoimmune diseases, indicative of the long-term influence of the execrated immune response excited during the infection (as is associated with ARDS). The linked autoimmune diseases include the Kawasaki disease, vasculitis, arthritis, hemolytic anemia, Guillain-Barré syndrome, and idiopathic inflammatory myopathies [21,22,23].

## 3. Management of ARDS

Currently, FDA-approved drugs against COVID-19 disease include remdesivir, which targets viral replication, and two other immunomodulatory drugs, namely, tocilizumab (IL-6 inhibitor) and baricitinib (JAK inhibitor) [24]. While the use of tocilizumab has shown a significant reduction in mortality in the treatment group (in several studies) when administered at early stages of infection [25], efficacy studies on baricitinib are not so conclusive [26]. Several studies have advocated for the use of immunomodulatory drugs against COVID-19 disease because of the important role ARDS plays in increasing disease severity by causing fatal local tissue damage. Hence, first, inhibitors of cytokines and cytokine receptors (monoclonal antibodies) have been seen as promising targets in stopping disease progression [10,27,28,29]. Other than cytokine inhibitors, dexamethasone, a drug with immunomodulatory properties, has demonstrated a reduction in the risk of mortality in patients on oxygen therapy. Similarly, another drug with anti-inflammatory properties, colchicine, has also shown significant benefits against disease progression, although further studies are recommended. Table 1 summarizes various drugs with immunomodulatory activities that can be used as an intervention against COVID-19 disease and have been tried clinically. Among the mentioned drugs, tocilizumab, tofacitinib, adalimumab, canakinumab, sarilumab, ravulizumab, baricitinib, and itolizumab act against the mediators known to be involved in the cytokine storm. Recognizing that the control of the disease is dependent both on the control of the viral replication and in controlling the dysregulated immune response, multiple combinations of drugs have been put forward, such as (a) remdesivir with baricitinib, (b) sirolimus and dactinomycin, (c) mercaptopurine and melatonin, (d) toremifene and emodin [30].

Although some of these drugs have shown positive results, the observations have been quite heterogeneous and hence call for more effective, universally applicable, or tailor-made treatment regimens against ARDS. In the following section, we provide details on different categories of drug candidates that are being used or can be considered for the discovery of novel/repurposed immunomodulatory drugs against ARDS associated with COVID-19. Please refer to Figure 5 for a schematic depicting the four classes of drugs discussed.

### 3.1. Anti-Rheumatic Drugs for COVID-19

Beyond the use of explicit anti-viral drugs, several medicines that often are used to treat rheumatic arthritis (RA) (with immunomodulatory properties) have been recommended as possible therapies for COVID-19 [56].

#### 3.1.1. Non-Steroidal Anti-Inflammatory Drugs (NSAIDs)

In RA, NSAIDs are routinely used in severe cases to alleviate joint inflammation. Several cohort studies have been conducted on the use of NSAIDs in COVID-19 and have concluded that no adverse events are linked with the use of NSAIDs during COVID-19 [57,58,59]. It is important to note that concrete evidence of the association of NSAIDs with long-term survival has also not been found, although NSAIDs have been previously used against other viral infections. In contrast, a study found no definitive evidence available to support the use of NSAIDs against COVID-19 and instead suggested NSAIDs might actually dampen antibody responses against the virus [60]. On the other hand, several other studies have concluded on continuing the course of NSAID treatment in patients with RA during the course of COVID-19 treatment [61]. The risk and beneficial effect of NSAID against local lung inflammation during ARDS needs further investigation.

#### 3.1.2. Corticosteroids

Corticosteroids are the basis for the control of infection flares and are used for RA treatment as verified by the most recently adopted the RA Board of Regulations of the European League for Rheumatism [62]. Corticosteroid treatment regimens have been associated with an increased risk of infections in patients as observed during observational exams. A collaborative study with over 15,000 RA patients over the age of 65 who used DMARDs (disease-modifying anti-rheumatic drugs) indicated glucocorticoids as the major risk factor for bacterial infections [63]. In addition, glucocorticoid use showed an increase in the progression of bacterial illnesses, while no increased risk of bacterial infection was observed in patients taking TNF inhibitors [64]. Moreover, the Global Registry information indicated that 33.31 percent of individuals who contracted COVID-19 diseases were on steroids, with 1783 rheumatism-contaminated patients. Their dispersed data indicated that the number of patients who received high doses of glucocorticoids (>10 mg/day) among those hospitalized was considerably greater than those who were not taking steroids [65].

To understand the effect of these drugs against COVID-19 disease, an investigational study was conducted on patients under DMARDs treatment that revealed an increased risk of disease severity in these patients upon baseline use. On the other hand, when administered in higher doses in patients suffering from severe COVID-19, these drugs showed positive outcomes [66]. Meanwhile, another study also advocated for the use of glucocorticoids during severe COVID-19, claiming that the drug suppresses IL-6 levels [67].

Given the ambiguity associated with the use of DMARDs, a paper reviewed the literature to gather a deeper understanding and concluded that the timing, duration, and amount of drug administration are key determinants of the outcome, claiming glucocorticoids to be “double-edged swords” [68]. A conclusion was reached about the use of these drugs in COVID-19 through the RECOVERY study. In the study, in patients on respiratory ventilation and oxygen, deaths were reduced by 28 days when patients were administered a reduced amount of dexamethasone (6 mg/day) for 10 days [69].

### 3.2. Natural Compounds as Immunomodulatory Agents

Natural compounds or the derivatives of traditional medicine from Asia are increasingly finding a place in both East and West with advanced studies being conducted to understand the intermolecular interactions of these compounds in the human body against specific diseases. The tools that were developed for understanding drug–target interaction on a structural and systemic level are finding their application in the excavation of a wealth of compounds available in traditional medicine—including pharmacodynamics, pharmacokinetics, and other pharmacology systems studies. Given this, anti-inflammatory and immunomodulatory compounds available from natural sources need detailed investigation. Colchicine and ginsenoside have been suggested as potent anti-inflammatory molecules and need further investigation for evaluating their effect on ARDS [70]. Similarly, in recent years, genistein has been increasingly reported to possess anti-inflammatory properties [71,72]. Interestingly, a molecular dynamics simulation study provided evidence of the effect of genistein on the catalytic activity of neutrophil elastase, which is known to be aggressively involved in neutrophil-mediated inflammatory action [73]. This finding warrants further investigation into the effect of this compound in ARDS management, given the integral role neutrophil plays in the syndrome.

Importantly, some natural compounds have already been studied for their effect on COVID-19. For example, Ali Nadi et al. (2023), through a literature survey, concluded on an immunosuppressive role of *Thymus vulgaris* and suggested that it acts by direct suppression of IL-6 and TNF-alpha and the induction of TGF-beta and IL-10. They proposed the use of the bio-actives derived from the plant in the management of COVID-19-related complications [74]. On a similar note, curcumin has also been suggested as a prophylactic agent against COVID-19 because of its anti-viral activities. It has been reported to target viral components, possess anti-inflammatory activities, and target IL-6 and HMG-B1, NK-kB associated TFs, and inflammasomes. The mentioned studies make curcumin, with its well-documented safety profile, a super candidate for ARDS prevention [75]. Furthermore, phytochemicals/bio-actives present in honey have also been reported to have a balanced immunomodulatory effect and need a more detailed investigation, specifically of their mechanism of action against ARDS [76]. A useful resource on this subject is the review written by researchers from China where they provide a detailed list of about 160 different flavonoids, alkaloids, terpenoids, polyphenols, quinonoid, and other naturally available bio-actives along with their potential molecular effects against different ARDS mediators. Table 2 provides a list of different natural compounds with potent anti-inflammatory activities along with their molecular details, which can be used for future studies on ARDS management. Although there seem to be several candidates that could have an immunomodulatory effect, the correct timing of usage along with the knowledge of the desired target at different time points seem to be essential factors to be considered to orchestrate an equilibrium [77].

### 3.3. Monoclonal Antibodies

Biologicals, specifically monoclonal antibodies (mAbs), have been used against several autoinflammatory/autoimmune diseases and cancers and have been seriously considered against ARDS in various clinical trials. The most used monoclonal antibodies directly target effector cytokines responsible for a hyperinflammatory response and include TNF-alpha inhibitors, IL1 receptor antagonists, IL2 receptor antagonists, and IL6 receptor antagonists. Moreover, nintedanib, a monoclonal antibody that acts as an inhibitor of tyrosine kinase, has already been licensed for chronic interstitial lung disease (ILD), as it stops the progression of fibrosis in lung tissue [78], while several other mAbs are under trial for use against ARDS to arrest acute local inflammation associated with COVID-19.

Table 1 provides a list of various drugs, including more than one anti-inflammatory monoclonal antibody under clinical trial to test immunomodulatory action during ARDS. Briefly, mAbs with the most promising results in multiple clinical trials were IL-6 inhibitors including tocilizumab, sarilumab, and siltuximab. mAbs against GM-CSF (Lenzilumab) and against IL-23 (Risankizumab) are also under clinical trials. IL-23 is increasingly reported to play an important role in local tissue inflammation [79,80,81] and hence is a target for subduing inflammatory responses. Leronlimab, a CCR5 antagonist, has also been shown to have an immunomodulatory effect in a clinical trial with 10 terminally ill COVID-19 patients and needs further clinical investigation [82]. Another important clinical trial (with 83 ARDS patients) on mAbs that is important to mention is of CERC-002, a neutralizing agent against TNFSF14, which proved to have significant activity in reducing mortality in the treatment group compared to the placebo group. Although the simplicity of action makes the use of these biologicals very attractive, the potential side effects and our inability to detect the current time of administration (based on disease progression) and to measure the bioavailability make the use of these agents quite challenging, highlighting the need for more research [83].

### 3.4. RNA Therapeutics

The RNA platform has recently been used as a vaccine where an RNA transcript is administrated through the vaccine formulation, which can (intracellularly) translate in the spike protein of SARS-CoV-2 to activate the adaptive immune responses [84,85]. This innovative vaccine platform is being further optimized and customized for its use in other infectious diseases [86]. RNA therapeutics, on the other hand, involves an interventional RNA targeting a gene/protein/transcript of interest for a particular disease to either downregulate (RNA interference) or degrade (with the use of anti-sense oligonucleotides (ASOs)) the molecule to cure or manage the disease. RNA aptamers can also be used to block proteins from their receptors [87]. Multiple RNA-based therapeutic products have been licensed for use in non-communicable diseases (for example, cardiovascular diseases and primary immunodeficiencies). For viral infections, RNA therapeutics under development mostly target viral proteins or mRNAs [88,89]. RNA therapies that aim for immune stimulation are also under development and mainly focus on anti-cancer treatments such as ASO 1018 ISS targeting TLR-9 for enhancement of cytotoxic effort function in non-Hodgkin’s lymphoma. Of specific interest to this article, RNA therapeutics are also under development that aim to harmonize hyperinflammation (again in cancer), such as aptamer NOX-A12 and NOX-E36, which disrupt the expression of CXCL12 and CCL2, respectively. If trials are deemed successful, the action of both the aptamers would disrupt myeloid infiltration in pancreatic cancer, colorectal cancer, and multiple myeloma (in the case of the former) and in diabetic nephropathy (in the case of the latter) [90].

The use of RNA-based interference for immunomodulation has also been advocated in bacterial infection-induced sepsis, where in one in vivo study of IRF-7 silencing was shown to arrest hyperinflammation-caused tissue damage in acute pyelonephritis and urosepsis [91]. More importantly, before the start of the COVID-19 pandemic, a review article briefed on the prospects of RNA therapeutics in acute respiratory distress syndrome. Along with discussing different kinds of RNA therapeutics that can be used against ARDS, the authors also indicated prospective targets, which included mediators involved in (i) reducing vascular permeability (Claudin-4, angiopoietin), (ii) reducing inflammation and cell death (MIP-2, Caspase-3, IL-6, HIPK1, S1P, Rip2, NF-kB, MTOR), and (iii) resolving inflammation and tissue repair (by the use of RNAi for upregulation of regulatory T cells (TF-foxp3), induction of IL-10 secretion by T cells). All the molecular mediators (potential targets of RNAi for ARDS) mentioned in the review were based on in vitro and in vivo experiments where the primary endpoint was the systemic reduction in the level of IL-6 and TNF-alpha. Clinical validation studies are still required to obtain the safety and efficacy profiles of these interventions [92]. Additionally, in vivo studies for acute myeloid leukemia (AML), cytokine release syndrome (CRS), and coxsackie virus B have suggested mir-146a as a potential therapeutic agent targeting the NF-kB-signaling pathway and the consecutive release of IL-1 and IL-6 inflammatory cytokines. The injected microRNA supposedly blocks TRAF-6 and IRAK-1 involved in the upstream regulation of the the NF-kB-signaling pathway [93,94]. Using transcriptomics data some studies have attempted to retrieve microRNAs that are associated with severe COVID-19 disease, which can be screened for their potential immunomodulatory properties [95,96].

As a novel approach, Cartesian Therapeutics (USA) developed a mesenchymal cell-based therapy called Descartes-30, which is powered by RNA-based cell engineering wherein the cells are armed with RNA therapeutics of interest. The therapy is being developed against ARDS by customizing mesenchymal cells in producing DNAse in greater quantities to target neutrophil extracellular traps (NETs), which play an instrumental role in the induction and progression of ARDs (as discussed above) [97].

Apart from cheap and easy manufacturing requirements, one of the prominent features of RNA-based therapeutics is the fact that they can be customized as needed using nucleotide synthesis facilities. Although the widespread use of these facilities in clinical settings is still in its infancy, one cannot restrain from speculating about the prospects of tailor-made immune-modulatory therapies in patients with various co-morbidities and for patients who are immunocompromised. Given this, RNA therapeutics could have significant implications in ARDS treatment and management [98]. Nevertheless, many challenges remain and two of the most prominent challenges in using RNA therapeutics against ARDS are selecting the right vehicle for administration [84,85] and bypassing the lung mucosa to target lung tissues with an exacerbated immune response [92]. Table 3 provides a list of key mediators of ARDS along with associated anti-sense RNAs that have therapeutic potential.

## 4. Advanced Computational Tools for Informed Drug Screening

The availability of large amounts of biological data and the enhancement of our ability to process computationally complex problems with a gain in computational power has facilitated several research studies on drug screening and discovery against multiple diseases, including SARS-CoV-2 [99]. Here, we discuss advanced algorithms that have been employed in this field in general, while exploring their applicability in finding immunomodulatory drug candidates that can be used against ARDS. It is to be noted that methods of structure-based drug discovery (where the target is known) were out of the scope of the current review and are not discussed in detail as we focus on both target and drug screening simultaneously.

### 4.1. Application of Network Medicine to Screen Immunomodulatory Drugs

Network-based studies of gene–gene and gene–drug interaction networks have been efficiently used to derive interactomes (virus–host) of interest. In one of the studies, omics data (transcriptomics data)) from the SARS-CoV-2 (severely) infected host was used to retrieve key genes/proteins and regulatory miRNAs using topological analysis of the formed network from the differentially expressed genes (DEGs). The genes were further validated by employing survival analysis for finding disease–gene associations. The obtained genes were further fed into DrugBank database, Therapeutic Target Database, and the Comparative Toxicogenomic Database to construct a disease–gene network. The formed network was validated using the STITCH database, a disease–protein database. The retrieved drug–target pairs were further tested using docking studies to propose promising drug candidates [95]. Using a similar approach, the authors screened out ERBB4 as a potential drug target along with Wortmannin as the drug. Here, differentially expressed genes were derived and then a protein–protein interaction network was constructed followed by the use of the Walkstrap algorithm to identify densely connected modules. The short-listed genes (from the modules) were subjected to docking studies to reveal potent drug–target pairs [100]. Another study integrated multi-omics datasets to retrieve differentially expressed genes in patients who succumbed to COVID-19 to develop a protein–protein interaction network and to screen out hub genes (genes that would play an integral role in COVID-19 progression) as potential drug targets [101].

In a different approach, the DEGs obtained from transcriptomics datasets (from both diseased and control samples) were used to generate a co-expression network using the STRING database before conducting clustering analysis to retrieve gene modules of interest. Thereafter, topological analysis was conducted to screen out hub genes using the MCODE plugin in Cytoscape, which were then used to retrieve the TG-TF-miRNA sub-network (Target gene, Transcription factor, and miRNA). These sub-networks were used as input for the Drug–Gene Interaction Database (DGIdb) to screen out drugs of interest that bind to TFs and non-TFs [96]. In another study, human transcriptomics data (and the enriched pathways) were instead used for validation of drug–target pairs obtained from protein–protein and drug–protein interaction networks. Specifically in this study, the objective to build a comprehensive protein–protein sub-network associated with responses to human coronavirus (not just SARS-CoV-2) was to churn out key drug targets [102].

In another research study specifically aimed at retrieving drugs that can be repurposed for COVID-19, the top 200 genes were used from a gene–gene interaction network that were known to interact with SARS-CoV-2 and were merged with drug–target networks (filtered for antineoplastic, immunomodulating, and anti-thrombotic properties) obtained from the DrugBank databases. The two networks (gene–gene and drug–target) were merged and topologically analyzed to determine potential drug targets. The network analyses were accompanied by functional enrichment, control path determination, and pathway analysis to delineate genes involved in pathways of interest. Based on the analysis, around 130 drugs were recovered with potential immunomodulating activities. The workflow followed in the research work provides a prototype for future research and can be further trimmed for sub-sections of the population with specific co-morbidities [103].

Another study followed a similar approach to using a drug–target network but instead of a gene–gene interaction network associated with COVID-19, the authors used a disease–gene interaction network for about a dozen diseases including rheumatoid arthritis, multiple sclerosis, cardiomyopathies, atherosclerosis, diabetes mellitus, HIV, malaria, SARS, and influenza. The diseases were chosen based on their genetic similarity to SARS-CoV2 (SARS), COVID-19-associated risk factors (diabetes), or similarity in the associated pathogenesis (viral infection—HIV). A novel algorithm named Searching off-label drug and network (SAveRUNNER) was used to merge the two networks to retrieve a drug–disease network. The algorithms included calculation of network proximity and similarity to the comprehensive human interactome, selection of drug–disease interactions “of significance” based on their proximity scores, followed by clustering analysis to segregate interactions with high modularity. The formed network was expanded and validated using networks available in the DrugBank and in Phenopedia. Additionally, new associations in the drug–disease network were calculated using a random walk procedure. As a validation step, the authors used gene set enrichment analysis (GSEA) wherein gene expression profiles (obtained from GEO) post SARS-CoV-2 infection were compared with the gene expression profile/signature associated with promising repurposed drugs (obtained from the Connectivity Map algorithm) derived through the SAveRUNNER algorithm. From the analysis, ACE inhibitors came out as highly potent drug candidates against COVID-19 along with immunomodulatory monoclonal antibodies (anti-IL6, anti-IL12, etc.) [104]. GraphVite, a high-performance system, was developed to find key association genes and drugs to reveal highly potent drug candidates [30]. Another sophisticated study used topological features from a drug–target network to extract proteins of high importance using machine learning algorithms that were verified using the Drug Bank [105].

Apart from static networks, dynamic networks have also been used to screen out immunomodulatory, anti-viral, and a combination of immunomodulatory–antiviral drugs. The basis of the study was the process of defining gene circuits (gene regulatory networks) that are robustly linked with the cytokine storm associated with the COVID-19 disease and, although incited by the virus, can remain activated long after viral clearance. Based on the defined circuits, key targets were chosen before screening and ranking of potential drug candidates [106].

### 4.2. Applications of Machine Learning Algorithms to Screen Immune-Modulatory Drugs

Machine learning has been used at different stages of drug discovery from target identification, compound screening, and compound property analysis, to the classification of clinical conditions for the optimal usage of treatment options and in the evolving branch of precision medicine. Both supervised (for example, regression models) and unsupervised (for example, clustering) machine learning algorithms have been used to predict target draggability (based on pharmacokinetic properties) along with the sequence and the structure information of the protein molecule. In addition, to predict the potential of prospective drugs, supervised learning can be used to develop models based on learning the properties of the registered or tested drugs [107].

Quantitate/Qualitative Structure Activity Relationship machine learning algorithms have been used to characterize chemical compounds with specific antimicrobial activities. For example, a research group developed a deep learning-based QSAR model to screen out chemical compounds with anti-parasitic activities along with low cytotoxicity in mammalian cell lines [108]. In QSAR, relationships between molecular properties and biological activities are characterized to screen out the most potent lead compound based on its molecular data. QSAR studies have been previously conducted to study compounds that bind to immunomodulatory receptors in vivo [109]. Similarly, the QSAR model has been developed to test cytotoxicity [110]. In fact, an ensemble model has been proposed wherein multiple models (majorly Random Forest (RF)) can be ingrained into a system to test for multiple different characteristics [111]. This approach can be a great asset in the discovery of immunomodulatory drugs that also do not have an immune-suppressive effect. Moreover, prospects of multi-target QSAR models allowing us to choose lead compounds that affect multiple targets are very exciting, given that a cytokine storm is a multifactorial phenomenon with multiple mediators involved [112].

Efforts in this direction have already begun; a sophisticated study used multi-target-based drug screening in which the researchers chose key inflammatory molecules from the literature by gauging their importance in the signaling pathways, specifically in the cytokine storm associated with COVID-19. These included the JAK1, STAT3, and NF-kB genes among other subcellular molecules. FDA-approved drugs against these molecules were screened out to develop a Random Forest (RF)-based predictive model that can access the inhibitory activity of the compounds being tested. The study produced remarkable results by retrieving drugs known to have inhibitory potential against multiple targets. The drugs with top inhibitory action (or top immune-modulatory activity) against most of the chosen targets were mitomycin C, abacavir, and raltegravir [113].

To predict drug properties post-administration, a supervised algorithm has been used to screen through the ADMET properties to select target-specific compounds that are well distributed and metabolized [114]. Furthermore, to monitor the effect of the drugs at the molecular level, gene expression-based ML models have been developed to set biomarkers indicative of drug action and efficiency [107].

Drugs against COVID-19 disease have already been screened out using ML approaches. For example, baricitinib, a Janus kinase inhibitor, was identified as a potent drug against COVID-19 by the BenevolentAI algorithm (machine learning algorithm) [115]. Similarly, dexamethasone, again with immunomodulatory properties, was identified by CoV-KGE, which is a deep learning framework that uses principles of network biology [116].

Aside from the norm, which aims for drug screening and identification directly, by using a novel methodology Saha et al. (2020) developed several machine learning algorithms that can predict drug targets against COVID-19 using therapeutic target databases, and they validated the gene acquired from the model with the available literature. The target genes that seemed novel were used to screen out repurposed drugs using the COVID19Db web server. The main highlight of the study was the use of unique properties as features to build the model: (i) protein–protein interaction features (which included a centrality score based on degree, closeness, between and other network parameters), (ii) Gene Set Enrichment Analysis (GSEA) ranking scores, and (iii) the properties of the amino acid sequences. This research work presented a demonstration of Machine Learning-based Drug Target Discovery (ML-DTD) [117].

#### Applications of Deep Learning

Deep learning is valued by biomedical researchers as it has the potential to capture and simulate biological data owing to its inherent architecture. Given this, several algorithms have also been used for drug discovery. A feedforward neural network developed by a group that classifies drugs based on the transcriptomic responses would be of particular interest in the search for immune-modulatory drugs. Apart from it, researchers have also used convolutional neural networks to find the binding affinities of the drugs to small molecules based on the available structural data. On the other hand, recurrent neural networks have been recommended to characterize binding properties based on protein sequence information [30,118].

On a similar note, in another study a graphical neural network was developed from a comprehensive and detailed gene-pathway-drug knowledge graph using different protein–protein interaction networks (which included genes from the virus and host responses). The drug–target interactions were retrieved from the toxicogenomic database (CTDbase). The generated knowledge graph was embedded in a pre-validated host-pathogen protein–protein interaction network to judge the accuracy of the knowledge graph and retrieve missing nodes from the network to attain completeness. The graphical neural network was trained to give binary outputs based on the usability of a particular drug against COVID-19 (based on drug involvement and performance in the clinical trial) [119]. This particular study is a notable attempt to combine the concepts of systems pharmacology with deep neural networks and for solving the “black box” problem, which is often labeled as a limitation of DL models. Moreover, the study also performed validation of the predicated drugs using gene expression datasets, results of in vitro studies, and conclusions derived from the analysis of the electronic health records of patients who received one or many of the drugs ranked high by the current DL model. The authors further elaborated the study by finding synergistic drug combinations based on the gene–gene interaction network associated with them and their respective phenotypes (effect in stopping disease progression). The same principle, though with a different methodology, was used in the development of the DeepCE model, which takes drug descriptors as input to predict expected gene expression profiles elicited by drug candidates [120]. Such models have important implications for understanding the mechanism of action of both novel and repurposed drugs, catalyzing the drug discovery process, and making regulatory decisions before market access.

To facilitate the reliable and productive application of machine learning algorithms in the process of drug discovery, TorchDrug, an extension built on PyTorch, was developed. The platform allows (i) molecules and their graphical representations to be processed and operated as regular datasets, (ii) the development of layered deep learning models, and (iii) the designing of de novo drugs based on the available knowledge of desirable molecular properties. Other tasks enabled in the platform include biomedical knowledge graph reasoning and molecular property prediction of the concerned drug molecule, given its molecular descriptors [121].

Although machine learning comes with a myriad of applications, they are still limited by some challenges, mainly because their applications in biology are still in their infancy and need further development specifically for customizable usage. The key challenges include (i) the lack of a systematic comparison capability of drug–target combinations to choose the superior pair, (ii) no standardized protocol for usage (which would aid the credibility and reproducibility of the application), (iii) lack of defined ML models to gauge toxicity, and (iv) ML/DL models being black boxes (where the internally detected patterns/models are difficult and often impossible to interpret) [122]. Availability of high-quality data, standardization of protocols involved in model development, and parameter optimization would enable screening out of highly credible drug–target pairs, which could have an immune-modulatory effect in specific disease conditions [107].

## 5. Discussion

The current study reviews four categories of drugs with immunomodulatory properties. Anti-rheumatic drugs are by nature immunosuppressors but need to be considered against COVID-19 with great caution as some of the classes might suppress the “protective” arm of the immune system, making the patient susceptible to other bacterial/fungal infections. Having said this, during severe infections, they can be life-savers, as reported in some studies. Monoclonal antibodies can also be very useful in severe infections, but persistent use has not been encouraged because of the side effects reported with their use in anti-cancer treatments. On the other hand, natural compounds have been suggested to act best as prophylactic drugs and can be recommended at the early stages of ARDS development. RNA therapeutic candidates, although fascinating, with their prospects being perceived as “quick and cheap” medications of the future, need thorough clinical investigation for detecting potential side effects and understanding their best possible usage against ARDS.

Moreover, the concept of precision medicine could be of vital importance in treating ARDS as effective activity of the administered immunomodulatory drug is dependent on the co-morbidities of the patient, the stage of the disease, along with the coronavirus variant under consideration (hence, a bucket of pre-validated drugs of different biochemical/medicinal properties would be of great value). Given this, work should be directed at using machine learning algorithms that recommend the drug or drug combinations that can effectively manage the disease in a patient- and situation-centric way. Considering that autoantibodies have also been found in some patients against important immunomodulatory cytokines, there is an imperative need to understand how pharmacogenomics studies can optimize interventions in patients pre-disposed to the production of certain autoantibodies [123]. Another avenue worth exploring is the immunomodulatory properties of natural compounds, which largely remain unexplored, and compare their activity and mechanism of action with effective FDA-approved drugs so that dependence on synthetically developed drugs can be reduced. Here, too, machine learning models along with the advanced characterization of the drug (natural compound)–target complex (structural biology) can play a pivotal role.

Currently, researchers have used gene–gene interaction networks, drug–gene networks, and human protein interactome networks for drug repurposing against COVID-19 in different combinations. As a step forward, researchers from Boston tried to develop a multi-modal algorithm that included the findings from different types of networks and approaches using a graphical convolutional neural network, which was characterized using network proximity analysis to find the close association between SARS-CoV-19-associated genes/protein and drug targets (not targets for immunomodulation). The drugs of interest obtained from the analysis were further validated by matching them with drugs under clinical trials and from the experimental evidence available in the literature. The validation and further screening of the drugs were conducted on the basis of their anti-viral activities. Acknowledging the requirement of tuning the innate immune response, along with attacking the virus, the authors suggested the potential of the same approach for screening and discovery of credible immunomodulatory drugs against COVID-19 [124].

Likewise, as we are currently in the exploratory stage of using ML applications for drug discovery and target prediction. Several groups are trying to optimize and refine the protocol in an innovative way. Here, a technical study on the use of autoencoders (neural networks) for feature selection is worth a mention. In this research work, the authors retrieved the SMILE IDs of both the drugs and the targets before converting them to their molecular formats. The molecular descriptors for the targets and for the drugs were derived from the Profeat tool and from the Rcpi library (R software), respectively, to create a co-aggregated matrix with features for both the target and the drugs. Using an autoencoder, the features were weighted and ranked in descending order (based on a weight coefficient) before being used as input for the construction of a multi-layer perceptron (MLP) model architecture to predict drug–target interaction. The model gave an impressive performance validating the devised innovative feature selection methodology. The entire methodology followed in this study and the highly ranked features (of drugs and targets) can be effectively used for finding reliable drug–target pairs with immunomodulatory effects against COVID-19 disease [125]. Here it is important to note that the current paper only discusses research studies where advanced network biology and machine learning-based algorithms were used for drug discovery against COVID-19. Although these approaches are very beneficial in understanding a drug’s mechanism of action, structural biology and drug–target binding studies would greatly increase the value of the computational pipeline being used in a drug discovery study.

Similarity and, as an advancement, dynamic network simulation studies can be a great asset for gathering a complete understanding of the immunomodulatory drug effect. After the construction of a static network, such studies involve the additional steps of defining a series of the ordinary differential equations (ODEs) called a system, which can represent the relation between effector elements (a cytokine/drug target, for example) with determining elements (mediator intracellular signaling pathways) based on assigned parameters [126,127]. If these models are trained from a heterogenous library of gene–gene and drug–gene interaction networks (from subjects with different ethnicity, co-morbidities, age, etc.), it would be possible to generate a robust all-encompassing model that would be able to predict drug action reliably for different parameter values (representing heterogeneous subject conditions). The development of such models would be a great leap in the quest for tools that could facilitate precision medicine. Identification and establishment of signature hyperinflammatory and immunomodulating gene regulatory networks and motifs (which can be considered as clubbed drug targets) in different host conditions at different time points of COVID-19 progression would be the first step in the dynamic network modeling [128,129,130,131] to predict precise drug action. Tools for retrieving such motifs are emerging and can be of great use [132].

Lastly, while considerable effort is being put forward to develop an efficient computational/bioinformatics pipeline for drug discovery from the R&D front, there is a need to make tools and applications for clinicians to aid the decision-making process during critical hours. Development of decision trees (after retrieving a bucket of pre-validated drugs), where the input would be the state of the disease in a subject (which could be represented by serum biomarkers, for example), that could suggest drug/drug combinations for use with high confidence scores would be an extraordinary development in providing relief against future flare-ups of COVID-19 disease.

## 6. Conclusions

Through this review, we characterized different categories of drugs under consideration against ARDS associated with COVID-19 disease along with discussing the current state of discovery of several candidates in each of the four categories. We also discussed the advancement of computational and systems/biology approaches in novel drug discovery and in drug repurposing. We found out that most of the studies focused on anti-virus drug discovery, particularly while COVID-19 disease severity and associated mortality are invariably related to ARDS. Hence, there is an urgent need to focus on in silico testing and validation of immunomodulatory drugs suitable and functional in different patients (with different ages, co-morbidities, genetics, etc.), which can be administered in combination with the anti-viral drugs being administered. Moreover, we noted multiple sophisticated studies which dealt with the application of machine learning in the process of drug discovery along with network biology approaches. Although these studies are fascinating at the current exploratory stage, there will soon be a need for standardized, reproducible algorithms to deliver high-quality, reliable drug candidates as the outcome of the in silico analysis.

## Figures and Tables

**Figure 1 jpm-13-00664-f001:**
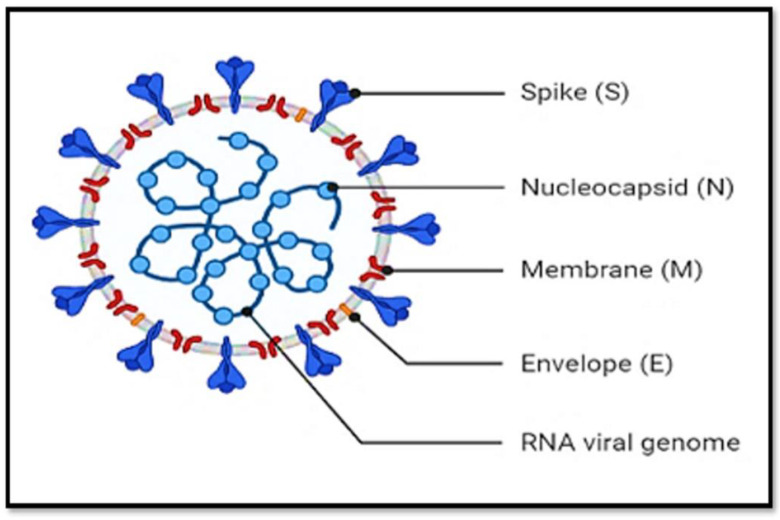
Structural representation of coronavirus 2019 (COVID-19).

**Figure 2 jpm-13-00664-f002:**
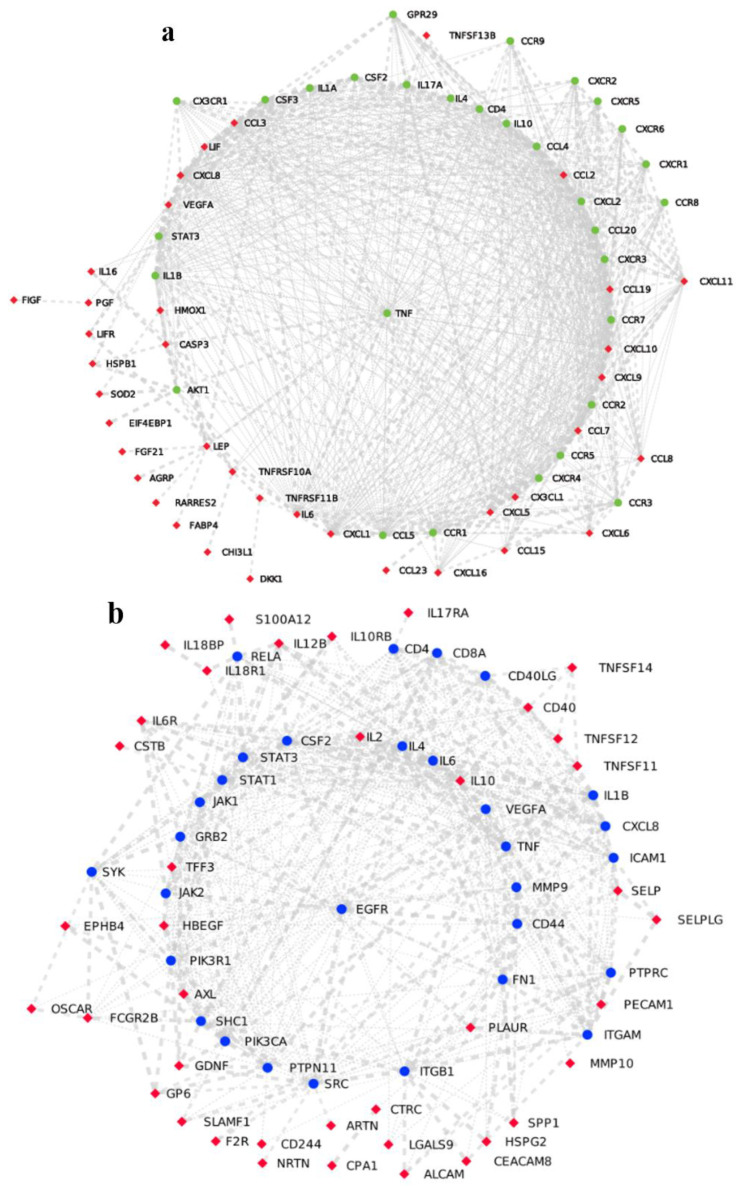
A protein–protein interaction network associated with (**a**) bacterial sepsis, where highly expressed genes are denoted in red and their interacting partners denoted in green and in (**b**) ARDS induced by COVID-19, where highly expressed genes are denoted in red and their interacting partners denoted in blue.

**Figure 3 jpm-13-00664-f003:**
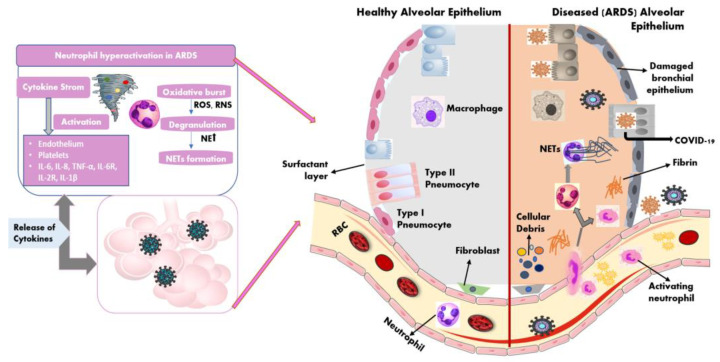
Neutrophils in lung tissue during SARS-CoV-2 infection and ARDS are depicted schematically. Neutrophil overexpression promotes cellular activities such as oxidation and cytokine production, as well as the production of multiple inflammatory mediators, including ROS, NE, NET formation, and various pro-inflammatory cytokines in an injured alveolar epithelium.

**Figure 4 jpm-13-00664-f004:**
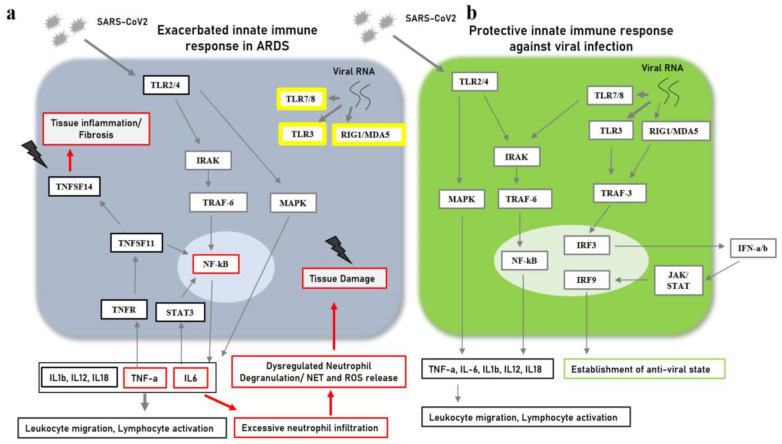
Signaling pathways associated with innate immune responses in the lung tissue (**a**) in ARDS and (**b**) during protective anti-viral immune responses.

**Figure 5 jpm-13-00664-f005:**
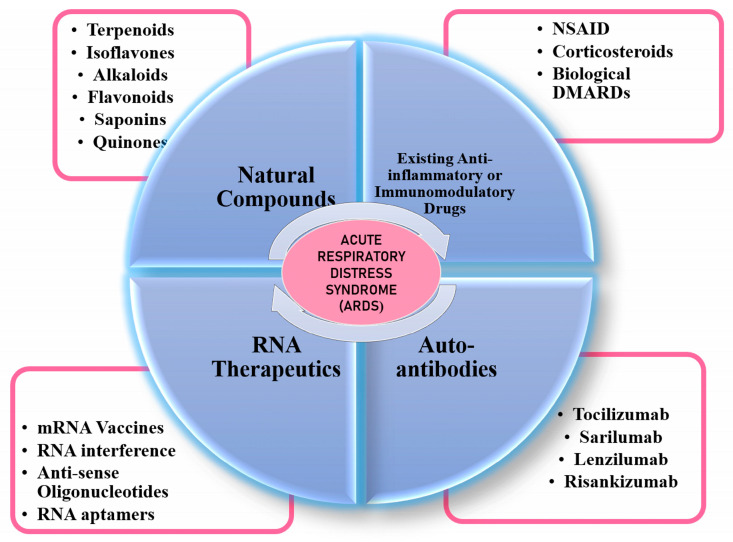
A schematic representation of different classes of immunomodulatory drugs that can be used against ARDS.

**Table 1 jpm-13-00664-t001:** List of immunomodulatory drugs/compounds clinically studied against COVID-19 disease.

S. No.	Drug Used	Current Pharmacological Indication	Drug Class/Function	Study Design	No. of Patients	Outcome	Ref
1	Ciclosporin and Favipiravir	Ciclosporin is used for the treatment of RA and Favipiravir used against influenza	Immuno-suppressant	Prospective, Non-controlled	20	No favorable effect detected	[31]
2	IFN-beta-1a along with Hydroxy-chloroquine and lopinavir/ritonavir	Pulmonary infections	Immuno-modulating	Prospective, Non-controlled	20	Significant improvement observed in patients	[32]
3	Pentoxifylline	Chronic occlusive arterial disease (COAD)	Rheologic modifier	Prospective, Controlled	38 (26T + 12C)	No statistically significant effect, but trend toward improvement in outcome parameters	[33]
4	Tocilizumab (subcutaneous)	Severely ill patients with COVID-19	mAb against IL6RA	Multi-center, prospective, open-label, uncontrolled study	126	Reduction in the risk of death when treatment begins in early stages of respiratory failure	[34]
5	Tocilizumab (subcutaneous)	Severely ill patients with COVID-19	,,	Open-label, multi-center, randomized, controlled, phase 3 trial (COVINTOC)	180 (90T, 90C)	No statistically significant reduction in disease progression, future studies advocated in severe patients	[35]
6	Tocilizumab	Severely ill patients with COVID-19	,,	Randomized prospective trial (COVIDSTORM)	84 (56T, 28C)	The intervention arm showed statistically significant clinical recovery and shortened stay in the hospital	[36]
7	Nanocurcumin	Liver cancer, colon cancer, and cancer of the central nervous system	Natural compound—Immunomodulatory	Prospective, controlled	120 (60T + 60C)	Statistically significant increase in the frequency and function of NK cells	[37]
8	Chlorpromazine	Schizophrenia	Typical antipsychotics	Pilot, multi-center, randomized, single blind, controlled, phase III therapeutic trial (standard arm vs. CPZ arm)	55T (in a cohort of 14,340 in-patients)	No statistical association between chlorpromazine administration and reduced mortality	[38,39]
9	Etanercept	Severely active RA and psoriatic arthritis	Tumor necrosis factor alpha inhibitor	Compendium of randomized controlled trials, transcriptional studies	5	Based on descriptive study showed therapeutic potential	[40]
10	Tofacitinib	RA and psoriatic arthritis	Janus kinase inhibitors	‘’	5	‘’	[40]
11	Adalimumab	Crohn’s disease and ankylosing spondylitis	DMARDs, tumor necrosis factor alpha inhibitor	‘’	3	‘’	[4]
12	Canakinumab	Autoinflammatory syndromes and systemic juvenile idiopathic arthritis (SJIA)	anti-IL-1beta monoclonal antibody	‘’	4	Potential effectiveness during high oxygen supplementation phase	[4]
13	Nintedanib	Pulmonary fibrosis and systemic sclerosis-associated lung disease	Tyrosine kinase inhibitors	Prospective, controlled study	30T, 30C	Statistically significant reduction in lung damage based on CT volumetry data (in patients with severe pneumonia induced by COVID-19)	[41]
14	Methylprednisolone	Neoplastic diseases and endocrine conditions	Glucocorticoids	Randomized, controlled study	76 (23T, 27C)	Statistically significant reduction in mortality and ICU admission in COVID-19 patients with severe pneumonia	[42]
15	Sarilumab	Moderate to severely affected patients with RA	mAB against IL-6 receptor	Randomized, double-blind, placebo-controlled, multinational phase 3 trial	416 (159T, 84C)	No statistically significant efficacy observed	[43]
16	RavulizumabBaricitinib		mAB against C5 and JAK, respectively	Randomized, parallel 3-arm (1:1:1 ratio), open-label, Phase IV—mulTi-Arm Therapeutic study in pre-ICU patients admitted with COVID-19 (TACTIC-R)			[44,45]
17	Targeted-synthetic/biological (ts/b) disease-modifying drugs (DMARDs)	Rheumatoid arthritis conditions	Anti-RA medication	Observational study on RA parients	2050	DMARDs administration do not put patients at increased risk	[46]
18	Corticosteroids	Used for treating chronic inflammations	Anti-inflammatory	,,	,,	Association with increased risk of COVID-19	[46]
19	Umbilical cord mesenchymal stem cell (UC)	Type 1 diabetes mellitus (T1DM) and T2DM, also gynecologic conditions	Stem cell supplement	Double-blind, phase 1/2a, randomized, controlled trial on subjects with COVID-19 induced ARDs	24 (12T, 12C)	Significant improvement in patient survival	[47]
20	Levamisole	Parasitic infections	Antiprotozoal agents, immunomodulatory effect	Prospective, double-blind, randomized controlled clinical trial	50 (50T, 50C)	Statistically improved cough status and dyspnea	[48]
21	High-dose Vitamin D	Hypoparathyroidism	Nutritional supplement	Multi-center, randomized, controlled, open-label, superiority trial	254 (127T, 127C)	Early administration of high-dose Vit. D improved overall mortality by day 14	[49]
22	Glycyrrhizin + Boswellic acids	Hyperglycemia and premenstrual syndromes	Natural compounds with anti-inflammatory and immunomodulatory properties	Single-center, randomized, double-blind, placebo-controlled, clinical trial	50, 25T, 25C	Statistically significant decrease in CRP and increase in lymphocyte level in intervention arm compared to control arm	[50]
23	Lianhuaqingwen	Influenza	Natural compounds with anti-inflammatory property	Prospective multi-center, open-label, randomized, controlled trial	284 (142T, 142C)	Significantly improved rate of recovery of symptoms and radiological reports	[51]
24	Nigella sativa	COVID-19	Natural compound with immunomodulatory properties	Open-label, randomized, controlled trial	183 (91T, 92C)	Significantly faster recovery of symptoms for mild COVID-19 infection	[52]
25	Itolizumab	Psoriasis	mAb against anti-CD46	Open, multi-center trial in elderly infected patients	19	Significant (up to 10 times) reduction of the risk of death	[53]
26	Itolizumab	Psoriasis	,,	ARDS paients	36	Significantly greater number of people had improved SpO2 level	[54]
27	Camostat mesylate	Psoriasis	Protease inhibitors	An open-label, phase I study to assess the safety, tolerability, and PK	14	The drug was well tolerated and the range of dosage was determined	[55]

**Table 2 jpm-13-00664-t002:** Immunomodulatory small molecules from the SuperNatural III database managed by Charite (Germany).

Sl. No.	Compound Name	Molecular Formula	Molecular Weight
1	8-METHOXY-PSORALEN	C_12_H_7_C_l_O_4_	250.63
2	9-HYDROXYELLIPTICINE	C_17_H_14_N_2_O	262.30
3	ACEMANNAN	C_66_H_100_NO_49_	1691.5
4	ANTHRAGALLOL	C_14_H_8_O_5_	256.21
5	ASIMICIN	C_37_H_66_O_7_	622.9
6	BAOHUOSIDE-1	C_27_H_30_O_11_	530.5
7	CHRYSAZIN	C_14_H_8_O_4_	240.21
8	EMODIN	C_15_H_10_O_5_	270.24
9	GALLIC-ACID	C_7_H_6_O_5_	170.12
10	GRAPHINONE	C_16_H_24_O_5_	296.36
11	HARMINE	C_13_H_12_N_2_O	212.25
12	LAPACHOL	C_15_H_14_O_3_	242.27
13	MATRINE	C_15_H_24_N_2_O	248.36
14	OSTHOL	C_15_H_16_O_3_	244.28
15	P-HYDROXY-BENZOIC-ACID	C_7_H_6_O_3_	138.12
16	PHORBOL	C_20_H_28_O_6_	364.4
17	POLYPHENOLS	C_20_H_22_O_9_	406.4
18	QUINIDINE	C_20_H_24_N_2_O_2_	324.4
19	SCOPARONE	C_11_H_10_O_4_	206.19
20	SESAMIN	C_20_H_18_O_6_	354.4
21	TANNIN	C_42_H_32_O_26_	952.7
22	MARINOL	C_21_H_30_O_2_	314.5
23	TETRANDRINE	C_38_H_42_N_2_O_6_	622.7
24	TYLOPHORINE	C_24_H_27_NO_4_	393.5
25	VANILLIC-ACID	C_8_H_8_O_4_	168.15
26	VANILLIN	C_8_H_8_O_3_	152.15
27	VERBASCOSIDE	C_29_H_36_O_15_	624.6
28	VINBLASTINE	C_46_H_58_N_4_O_9_	811.0
29	VINCRISTINE	C_46_H_56_N_4_O_10_	825.0
30	WITHAFERIN-A	C_28_H_38_O_6_	470.6
31	WITHANOLIDE-D	C_28_H_38_O_6_	470.6
32	(+)−EPIPINORESINOL	C_20_H_22_O_6_	358.4
33	3-ACETYLACONITINE	C_36_H_49_NO_12_	687.8
34	ACONITINE	C_34_H_47_NO_11_	645.7
35	ADENOSINE	C_10_H_13_N_5_O_4_	267.24
36	ALKANNIN	C_16_H_16_O_5_	288.29
37	ALPHA-TOCOPHEROL	C_29_H_50_O_2_	430.7
38	ARCTIGENIN	C_21_H_24_O_6_	372.4
39	ARTEMISININ	C_15_H_22_O_5_	282.33
40	ASCORBIC-ACID	C_6_H_8_O_6_	176.12
41	BOLDINE	C_19_H_21_NO_4_	327.4
42	CHIMAPHYLIN	C_12_H_10_O_2_	186.21
43	EUCOMMIN-A	C_27_H_34_O_12_	550.6
44	GAMMA-LINOLENIC-ACID	C_18_H_30_O_2_	278.4
45	GINKGOLIDE	C_20_H_24_O_10_	424.4
46	INOSINE	C_10_H_12_N_4_O_5_	268.23
47	IRILONE	C_16_H_10_O_6_	298.25
48	LIMONENE	C_10_H_16_	136.23
49	LINOLEIC-ACID	C_18_H_32_O_2_	280.4
50	OLEANOLIC-ACID	C_30_H_48_O_3_	456.7
51	PAEONOL	C_9_H_10_O_3_	166.17
52	ROSMARINIC-ACID	C_18_H_16_O_8_	360.3
53	RUTIN	C_27_H_30_O_16_	610.5
54	SAIKOSAPONIN	C_42_H_68_O_13_	781.0
55	SAPONINS	C_55_H_86_O_24_	1131.3
56	SWAINSONINE	C_8_H_15_NO_3_	173.21
57	SYRINGIN	C_17_H_24_O_9_	372.4
58	TOCOPHEROL	C_29_H_50_O_2_	430.7
59	URSOLIC-ACID	C_30_H_48_O_3_	456.7
60	WITHANOLIDE	C_28_H_38_O_6_	470.6

“https://bioinf-applied.charite.de/supernatural_3/index.php (accessed on 11 February 2023)”.

**Table 3 jpm-13-00664-t003:** Inflammatory mediators involved in ARDS and associated anti-sense RNA as reported in the NCBI database.

Sl. No.	Inflammatory Mediator	Classification	Anti-Sense RNAs
1	IL-1β	Pro-inflammatory	NA
2	IL-6	Pro-inflammatory	IL6-AS1
3	IL-18	Pro-inflammatory	NA
4	IL-1	Pro-inflammatory	NA
5	TNFα	Pro-inflammatory	HOTAIR, KCNK15-AS1
6	IL-10	Anti-inflammatory	GNAS-AS1
7	TGF-β	Anti-inflammatory	CDKN2B-AS1, KCNQ1OT1, AFAP1-AS1, AFAP1-AS1, NNT-AS1, WT1-AS, HAS2-AS1, HAS2-AS1, MBNL1-AS1, PRR34-AS1, TGFB2-AS1
8	ELANE	Innate immunity	NA
9	CXCR2	Chemokines	NA
10	MMP9	Chemokines	SLC12A5-AS1, TP73-AS1, HAGLR
11	CXCL2	Chemokines	NA
12	IFN-γ	Pro-inflammatory	IFNG-AS1
13	PAF	Signaling pathway (intercellular mediator)	NA
14	GM-CSF	Adaptive immune system	NA
15	C5a	Pro-inflammatory	NA
16	ICAM-1	Neutrophil adhesion	LIMASI, ICAM4-AS1
17	VEGF	Endothelial cytokine	HOTAIR
18	IGF-I	Alveolar macrophage	HOXA-AS2
19	ROS	Regulation of vascular tone	NA
20	NLRP3	Extracellular histone	HOTAIR, CDKN2B-AS1, HAGLR, DLX6-AS1, ADAMTS9-AS2, RGMB-AS1, ZNF561-AS1
21	IL-2	Adaptive immunity	NA
22	IL-37(IL-1F7)	Anti-inflammatory	NA
23	TLR4	Extracellular histones	PAPPA-AS1, MGAT3-AS1
24	CXCL10	Cytokines	NA
25	M-CSF	Pro-inflammatory	NA
26	NF-κΒ	Pro-inflammatory	SLC26A4-AS1
27	MIP-1α	Chemokine	NA
28	MIP-1β	Chemokine	NA
29	IL-6R	Pro-inflammatory	IL6R-AS1
30	CXCL9	Monokine	NA

## Data Availability

No new data was created for this article. Secondary dataset from another work (https://doi.org/10.1186/S10020-023-00609-6) was used for the construction of PPI networks present in Figure 2.

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
