# Peer review of "Prospects of Novel and Repurposed Immunomodulatory Drugs against Acute Respiratory Distress Syndrome (ARDS) Associated with COVID-19 Disease"

_jpm, 2023, doi:10.3390/jpm13040664_

Round 1

Reviewer 1 Report

this is a lengthy but very interesting review on the repurposing for covid 19 which is going to be very useful for the readers"

my only minor comments

if you have evidence or based on your knowledge please mention which of these compounds is appropriate for citokine storm apart from corticosteroids or tocilizumab

also corticosteroids cannot be considered as repurposing because they are used in ARDS and in any viral pneumonia where there is a need (eg high systemic inflammation, respiratory failure)

Author Response

Reviewer 1

This is a lengthy but very interesting review on the repurposing for covid 19 which is going to be very useful for the readers". My only minor comments

Comment#1 If you have evidence or based on your knowledge please mention which of these compounds is appropriate for cytokine storm apart from corticosteroids or tocilizumab

Response: The suggested material has been incorporated and highlighted in yellow in Table 1

Comment#2 Also corticosteroids cannot be considered as repurposing because they are used in ARDS and in any viral pneumonia where there is a need (eg high systemic inflammation, respiratory failure)

Response: As per the reviewers’ suggestion the main title of the concerned section has been changed to “Anti-rheumatic drugs against COVID-19”. Moreover, section 3.1.2 discusses in detail about the lack of a clear picture of treatment outcomes in COVID-19 while also pointing towards the conventional use of corticosteroids in other viral infections. The mentioned changes are highlighted in yellow.

Reviewer 2 Report

General comment

The authors propose a review article that summarizes therapeutic opportunities under consideration against ARDS associated with COVID-19, discussing different drug categories. The addresses a relevant topic to the scientific community, but a review article should be more detailed and deepened about the investigated topic. Thus, some improvements and clarifications need to be performed to the presentation of the study, especially by adding also some computational tools for drug repurposing.

Major Comments

1.       The study described in the manuscript is really interesting, but the authors should better emphasize the significant advance of this study within the field also of drug repurposing, pointing out repurposing attempts for the understudied drugs. In this regard, the authors should cite also approaches for in-silico drug repurposing used for predicting repurposable COVID-19 solutions (some of them discussed also some of the cited drugs such as tocilizumab):

1)      Network medicine framework for identifying drug-repurposing opportunities for COVID-19 PNAS May 11, 2021 118 (19) e2025581118; https://doi.org/10.1073/pnas.2025581118

2)      SAveRUNNER: A network-based algorithm for drug repurposing and its application to COVID-19. PLoS Comput Biol 2021 17(2): e1008686. https://doi.org/10.1371/journal.pcbi.1008686

2.       Following the indication about repurposing, in my opinion, Table 1 should present also a column with the original medical indications of the listed drugs.

3.       The authors should add a graphical abstract to help the reader to focus of the review topic and main findings.

4.       The figures should be improved. Most of them have fonts that are barely readable (e.g., Fig2) or are stretched (e.g., Fig 1)

Minor

1.       In general, the authors should check the form of the entire paper avoiding careless errors, missing commas, missing space between words.

2.       In general, the authors should check the English language and the form of the entire paper avoiding careless errors, missing commas and dots, missing space between words, and confusing sentences that should be reworded in order to make them clearer. All the writing part needs to be thoroughly checked and improved throughout the manuscript to bring it to an acceptable level. I suggest to submit the manuscript to a native-speaker reading or to English editing program.

3.       Check all the commas. Add a comma before “and” whenever more than two elements have been listed and before “while”.

Author Response

Reviewer 2

General comment

The authors propose a review article that summarizes therapeutic opportunities under consideration against ARDS associated with COVID-19, discussing different drug categories. The addresses a relevant topic to the scientific community, but a review article should be more detailed and deepened about the investigated topic. Thus, some improvements and clarifications need to be performed to the presentation of the study, especially by adding also some computational tools for drug repurposing.

Major Comments

Comment#1 The study described in the manuscript is really interesting, but the authors should better emphasize the significant advance of this study within the field also of drug repurposing, pointing out repurposing attempts for the understudied drugs. In this regard, the authors should cite also approaches for in-silico drug repurposing used for predicting repurposable COVID-19 solutions (some of them discussed also some of the cited drugs such as tocilizumab):

Network medicine framework for identifying drug-repurposing opportunities for COVID-19 PNAS May 11, 2021 118 (19) e2025581118; https://doi.org/10.1073/pnas.2025581118

SAveRUNNER: A network-based algorithm for drug repurposing and its application to COVID-19. PLoS Comput Biol 2021 17(2): e1008686. https://doi.org/10.1371/journal.pcbi.1008686

Response: Based on the kind suggestion of the reviewer, in silico studies with COVID-19 as the focus have been included in the review article and those bits have been highlighted in section 4.1 and 4.2 and in the reference section

Comment#2       Following the indication about repurposing, in my opinion, Table 1 should present also a column with the original medical indications of the listed drugs.

Response: Original medical indications have been added to the Table 1

Comment#3     The authors should add a graphical abstract to help the reader to focus of the review topic and main findings.

Response: Upon the editor’s suggestion earlier, a graphical abstract has been created and submitted later. We include the GA in page#2 of the updated manuscript. 

Comment#4      The figures should be improved. Most of them have fonts that are barely readable (e.g., Fig2) or are stretched (e.g., Fig 1)

Response: Same figures with better quality have been inserted replacing the previous ones

Minor

Comment#1     In general, the authors should check the form of the entire paper avoiding careless errors, missing commas, missing space between words.

Response: Significant time have been spent in checking the quality of the manuscript in terms of spacing, format, language and punctuations. The changes have been done.

Comment#2     In general, the authors should check the English language and the form of the entire paper avoiding careless errors, missing commas and dots, missing space between words, and confusing sentences that should be reworded in order to make them clearer. All the writing part needs to be thoroughly checked and improved throughout the manuscript to bring it to an acceptable level. I suggest to submit the manuscript to a native-speaker reading or to English editing program.

 Response: As per reviewer’s kind suggestion significant time has been spent in checking the quality of the manuscript in terms of spacing, format, language and punctuation. The manuscript has also been re-read and corrected by proficient English speakers.

Comment#3    Check all the commas. Add a comma before “and” whenever more than two elements have been listed and before “while”.

Response: The suggested changes have been made in the manuscript

Round 2

Reviewer 2 Report

I appreciated the effort of authors in addressing all my issues risen in the previous round of revisions, thus I suggest to accept the manuscript